# Effects of Message Framing, Sender Authority, and Recipients’ Self-Reported Trait Autonomy on Endorsement of Health and Safety Measures during the Early COVID-19 Pandemic

**DOI:** 10.3390/ijerph18157740

**Published:** 2021-07-21

**Authors:** Elli Zey, Sabine Windmann

**Affiliations:** Institute for Psychology, General Psychology II, Goethe University Frankfurt, 60323 Frankfurt am Main, Germany; s.windmann@psych.uni-frankfurt.de

**Keywords:** autonomy, morality, authority, prosocial behavior, framing, messaging, COVID-19 regulations, social distancing

## Abstract

In the COVID-19 pandemic, human solidarity plays a crucial role in meeting this maybe greatest modern societal challenge. Public health communication targets enhancing collective compliance with protective health and safety regulations. Here, we asked whether authoritarian/controlling message framing as compared to a neutral message framing may be more effective than moralizing/prosocial message framing and whether recipients’ self-rated trait autonomy might lessen these effects. In a German sample (*n* = 708), we measured approval of seven regulations (e.g., reducing contact, wearing a mask) before and after presenting one of three Twitter messages (authoritarian, moralizing, neutral/control) presented by either a high-authority sender (state secretary) or a low-authority sender (social worker). We found that overall, the messages successfully increased participants’ endorsement of the regulations, but only weakly so because of ceiling effects. Highly autonomous participants showed more consistent responses across the two measurements, i.e., lower response shifting, in line with the concept of reactive autonomy. Specifically, when the sender was a social worker, response shifting correlated negatively with trait autonomy. We suggest that a trusted sender encourages more variable responses to imposed societal regulations in individuals low in autonomy, and we discuss several aspects that may improve health communication.

## 1. Introduction

Social media can have a profound impact on how we understand our societies, what we anticipate and experience, what we value, how we feel, and how we behave. In order to convince people to engage in a certain behavior, what matters is not only the content of the message, but also how and by whom it is delivered.

### 1.1. Literature Review

Regarding the how, *message framing* is one way to vary the persuasiveness of delivered information [1]. First, Tversky and Kahnemann looked into the phenomenon of why people systematically violate consistency and coherence in rational decision-making, and they demonstrated that seemingly inconsequential changes in the formulation (framing) of choice problems caused large and systematic shifts of choice preferences even though mathematically, the expectancy value of all options remained the same. In the original research, most often loss and gain framing have been compared. More generally, Tversky and Kahnemann describe three different types of framing: the framing of acts, contingencies, and outcomes, and the characteristic nonlinearity of values and decision weights [2].

Since then, many empirical studies have confirmed that message framing in communication has a significant effect on judgment and decision making [3], extending to the domain of health protection behaviors [4]. For example, short reminders sent via smartphone have been shown to increase adherence to drug treatment plans [5]. With the right framing, smartphone messages can also function as a reminder to act morally “good” and for the well-being of others. Prosocial framings that highlight the role of others, such as close persons, one’s children or loved ones, and even strangers, have been shown to increase people’s intentions to get a vaccination, more so than a self-oriented frame did [6].

By contrast, a binding moral frame was found to effectively shift decisions of conservative participants into a pro-environmental direction when protecting the environment was framed as a matter of obeying authority [7]. Especially in times of threat to collective well-being as through a global pandemic, the prevailing feeling of uncertainty might make groups or societies susceptible to authoritarianism [8,9,10]. Under those conditions, strong injunctive norms might provide a feeling of safety with regards to how one should or should not behave [11]; thus, demanding or controlling message framing might be most effective. 

Sender characteristics such as their authority status may be other key factors when it comes to the question of effective communication. In his famous experiment, Milgram investigated the decisions of participants under the influence of an authority figure, the experimenter, and found a very high proportion of participants to give electric shocks to another person merely because the experimenter told them to do so [11]. More recent research confirms that an authority or legal system, when perceived as legitimate, does not require any type of explanation or justification for people to obey [12,13]. It appears as if people tend to succumb to the influence of leadership once they accept the existing power relations. 

However, not all people are the same, in the sense that message framing and sender characteristics are likely to interact with the personality traits of the recipients of the message. The arguably most important personality trait in this context may be *autonomy* (Greek αuτόνομος: ‘auto’ means self and ‘nomos’ means law), literally translated best as the ability to follow one’s own rules. According to Piaget, an individual is autonomous if decisions and actions are independent of external influences, especially of adult authority [14]. Other developmental scientists also associate autonomy with not conforming to others, or not reacting to social judgment, again especially that of adults [15,16]. Such conceptualizations are captured by the term *reactive autonomy* [17]. Modern frameworks see autonomy more proactively linked to an agent’s ability to determine and shape their own environment [18]. The present study compromises in understanding autonomy as consistently self-determined thinking and acting, which implies (but is not limited to) resistance against social influences. We set out to investigate: how do high- as compared to low-autonomy individuals respond to demanding regulatory messages in times of the COVID-19 crisis?

### 1.2. Global SARS-CoV-2 Pandemic

The global SARS-CoV-2 (severe acute respiratory syndrome coronavirus 2) pandemic has been and still is threatening the health and lives of millions of people. To reduce the transmission of the virus and the spread of the disease, several health measures (e.g., physical distancing, quarantine, and handwashing) were ordered by a number of governments and authorities since the outbreak. Compliance of individuals with these measures is essential to slow down the spread of the virus [19]. Thus, the situation requires each and every individual to accept restrictions on their personal freedom and autonomy, for the greater good of all. 

Many countries around the world implemented a number of nonpharmaceutical interventions colloquially known as lock-downs (encompassing stay-at-home orders, curfews, quarantines, and other regulations) to reduce the spread of SARS-CoV-2 which causes COVID-19 [20]. As in most Western, educated, industrialized, rich, and democratic (WEIRD) countries, a stay-at-home order was also instituted in Germany in early April 2020, the time and place where the present study was conducted. During this time, the public was asked to stay at home; only so-called system-relevant branches were allowed to work outside of a home office; universities, schools, and kinder-gardens were closed; and people were advised to reduce their contacts to an absolute minimum and were allowed to meet with only one further person of a different household in public.

### 1.3. Pandemic Situation in Germany at the Time of the Study

We used this early pandemic situation in Germany as our paradigm: We asked citizens to report their compliance with currently imposed behavioral protection measures and investigated whether advice by a person of high authority (state secretary) versus low authority (social worker) in the sense of hierarchical leadership would increase endorsement of the behaviors. In addition, we varied the framing of the messages conveyed by the advisors: The authoritarian message argued with the law and referred to executive enforcement measures by the police, whereas the moralistic message argued with one’s own responsibility for the community and oneself. A neutral control message with no particular framing was also included for reasons of comparison. We hypothesized that the authoritarian message would be most effective in influencing self-reported compliant behaviors, especially if sent by the high-authority figure.

Importantly, we determined participants’ trait autonomy by established self-reported questionnaire items, and we asked whether it would interact with the experimental interventions. In line with the reactive component of our concept of autonomy, we predicted that individuals high in trait autonomy would show more consistent responses before and after reading the message; that is, they would resist the influences of the messages more than those low in trait autonomy. Hence autonomy should correlate negatively with the shifting (pre-post difference) in the responses due to the experimental interventions. We further predicted that this resistance against change would be higher (i.e., correlation less negative) for the authoritarian message sent by the high-status sender compared to that sent by the low-status sender because we envision autonomy to be directed not primarily against change in principle, but primarily against change imposed by powerful forces. 

## 2. Materials and Methods

### 2.1. Participants

The survey was conducted in the early days of the COVID-19 pandemic response in Germany from 16 April 2020 to 20 April 2020. Participants were recruited either in collaboration with the panel Consumerfieldwork (http://www.consumerfieldwork.de (accessed on 19 July 2021)), *n* = 300, or via personal inquiries and social media. One hundred five students of Psychology at Goethe University Frankfurt participated for course credits. Panelists were rewarded according to their compensation agreement with Consumerfieldwork (*M =* EUR 0.80). We excluded participants who did not complete the whole survey (*n* = 202). All of these answered the two attention check questions (e.g., *“This is a question for attention control. Please check the second box from the left.”*) correctly. We further excluded participants, who reported being under 18 or over 120 years (*n* = 3) of age, or who completed the survey in less than the median participation time multiplied by 0.25 (*n* = 1). The final data set consisted of *n* = 707 participants (454 female, no diverse), who finished the survey in *M =* 10.4 min. Age ranged from 18 to 85 years; *M =* 37.56 (*SD* = 17.75). Participants reported no school leaving degree (one person), school leaving certificate (5%), secondary school leaving certificate (15%), A-levels (37%), trained profession (20%), or university/college degree (23%) as highest achieved educational degree. 

### 2.2. Design, Procedure, and Measures

The design was a mixed factorial design involving within-participants effects (pre- vs. post-intervention) and the between-participants factors leadership status (high for *state secretary*, *n* = 354, and low for *social worker*, *n* =353) and message framing (authoritarian, *n* = 233, vs. moral, *n* = 238, vs. none, control, *n* = 236). Participants were randomly assigned to one of the six between-factor groups. 

The experiment was performed online. In the pre-intervention measurement, seven behavioral items were presented about social distancing behaviors in accordance with current governmental regulations in Germany at the time (see Table 1). Next, participants answered 22 items assessing autonomy as a personality trait, chosen to reflect our conception of autonomy as consistent responding despite social influences (McDonald’s *ω =* 0.81, see Appendix A for list of items). Of these, 10 items were taken from the Moral Agency Scale [21], e.g., “*In most cases, I can make my own decisions about what is right or wrong in a situation*”; 6 items were adapted from the Trier Personality Questionnaire [22], e.g., “*I like to go my own way*”; and a further 6 items were adapted from the protective social comparison scale [23], e.g., inverted item “*My behavior often depends on how I feel others wish me to behave*”. Thereafter, the experimental treatment was provided. Participants were shown a Twitter post (the post was fictitious, but this was unknown to them) that varied between groups of participants by sender and message framing (Figure 1). Next, a memory check was administered. One was a multiple-choice question asking about the occupation of the sender of the post, and the other asked about the reasoning used in the message. Because only 479 of the 707 answered both these items correctly, we refrained from excluding any of the participants based on this check. The participants also rated the senders’ trustworthiness and the senders’ morality. They were then given the seven items on the social distancing behaviors again in the post-treatment measurement. We also displayed the moral and the authoritarian message to the participants and asked for the effectiveness of the two messages. Finally, we asked participants five questions rated on a 5-point-Likert-scale: how much they felt the pandemic to be a threat for society, for themselves personally, and for their close social environment and how they evaluated their personal risk and the risk to their close social environment. We also assessed whether the participants themselves or someone in their households had tested positive for COVID-19 or people in their direct environment had tested positive. At the end, after answering demographic questions about their person, participants were thanked and debriefed. All participants provided informed consent, and the study was approved by the institutional ethics board of our faculty. 

### 2.3. Statistical Analyses

The mean ratings of the seven behavioral safety measures taken before and after the manipulation and the ratings of the 22 autonomy items were computed. We computed the pre-post difference by subtracting the post-intervention value from the pre-intervention value for each item and each person. For analyses involving trait autonomy, we used the absolute pre- and post-intervention values because our hypotheses referred to the extent of the shifting between pre- and post-intervention measurement, not the direction of the shift. Inferential statistical analyses were performed in line with the preregistration [24] as follows: 

*Analysis 1.* We conducted an ANOVA on the average responses across all 7 pre-intervention measurement and post-intervention measurement questions. This was a 2 (pre-post) × 2 (author (=sender)) × 3 (message framing) factorial design [24]. However, because this analysis yielded no significant experimental effects other than a significant pre-post difference (see Section 3), presumably due to ceiling effects on many of the items, we inspected the effects at the level of single items and found that Item 5 did not show such ceiling effects. We, therefore, analyzed responses to Item 5 separately using the same ANOVA. 

*Analysis 2*. We correlated the average score of the 22 trait autonomy questions with the pre-post difference across all questions in all 6 groups. Additionally, we analyzed the pre-post difference by multiple regression analyses using trait autonomy and authoritarian treatment (leadership status (=sender) and message framing) as a predictor [24]. 

*Analysis 3*. “To rule out floor/ceiling effects (response rates below 0.20 or above 0.80), we will repeat the analyses using only items with response rates between 0.20 and 0.80 (averaged across all groups)”, quoted from [24]. There is a mistake in the wording of the dependent variable in this section: It refers to “response rates” where it should refer to “ratings”. Because we did indeed find a reason to suspect the presence of ceiling effects, we dropped ratings above 0.80 of the Likert scale in the pre-treatment measure (i.e., values that were already maximal to begin with), then calculated the pre-post differences of each participant using only the remaining items, and repeated the ANOVA described in analyses 1 and 2. 

*Explorative Analysis.* Social demographic values are assessed in the Appendix A in Appendix A.

Analyses were performed with the programming language R-4.1.0, using RStudio (version 1.4.1106); the significance level was set to *p* = 0.05.

## 3. Results

### 3.1. Manipulation Check and Descriptive Results 

Participants rated the senders’ trustworthiness in the state secretary group (*M =* 3.34, *SD =* 0.97) significantly lower than that in the group with the social worker as sender (*M =* 3.45, *SD* = 0.92; *t*(703.28) = −1.65, *p* < 0.049). The same effect was found for morality: the state secretary (*M =* 3.63, *SD* = 0.87) was rated significantly less moral compared to the social worker *(M =* 3.96, *SD* = 0.81; *t*(701.33) = −5.15, *p* < 0.01). Participants rated the moral/prosocial message (*M =* 4.27, *SD* = 0.91) as significantly more effective than the authoritarian/controlling message (*M =* 3.14, *SD* = 1.29; *t*(706) = −19.81, *p* < 0.01). 

On average, participants reported the pandemic to be more of a threat for society (*M =* 3.80, *SD* = 0.93) and for their close social environment (*M =* 3.74, *SD* = 1.11) than for themselves personally (*M =* 2.65, *SD* = 1.30). The difference between personal threat and societal threat was significant (*t*(706) = −24.36, *p* < 0.01), as was the difference between personal threat and threat for the close social environment (*t*(706) = −22.26, *p* < 0.01). Furthermore, participants perceived themselves much less as part of a high-risk group (*M =* 2.19, *SD* = 1.47) than they did the people in their households (*M =* 2.73, *SD* = 1.61; *t*(706) = −8.77, *p* < 0.01).

In their direct environment, 104 participants reported positive cases, 522 participants reported no positive cases, and 81 reported being uncertain. Four participants had tested positive, four more reported persons testing positive in their households. 43 participants reported symptoms but had not been tested, and this was the case for 15 persons in the households of participants. 641 people reported no symptoms or positive tests for themselves or their households since the start of the pandemic.

### 3.2. Main Analyses

*Analysis 1.* The main ANOVA found a small but significant effect of the repeated measures factor. The average pre-intervention rating across all seven items (*M =* 4.07, *SD* = 0.68) was significantly lower than the post-intervention rating across all seven items (*M =* 4.14, *SD* = 0.71; *F*(1, 701) = 19.55, *p* < 0.001, *η^2^* = 0.002). No other effects were significant.

Exploratory Analyses. Average pre-intervention rating values were above 4; in fact, 2768 out of 4949 pre-intervention ratings (56%) were at the maximum value of 5 to begin with. Therefore, we inspected results at the single-item level and noticed that Item 5 was the only one that was far away from showing such ceiling effects. The item asked about wearing a mask in public indoor spaces, a measure that was not common at the time of the survey and was in fact not officially recommended yet. We thus explored the effects of our experimental manipulations on this item alone. As shown in Figure 2, participants endorsed wearing a mask in public indoor spaces much more after the intervention (*M* = 2.79, 95% CI [2.68, 2.91]) than before (*M* = 2.23, 95% CI [2.12, 2.34]; *F*(1, 701) = 220.662, *p* = 0.03, *η^2^* = 0.034). No other effects were significant. The sender x message interaction was at *F*(2, 701) = 1.144, *p* < 0.32, *η^2^* = 0.003. Results and graphs for the other items are shown in Appendix A and Figure 1 and Figure 2.

*Analysis* 2. Across all seven items, bidirectional pre-post differences did not correlate significantly with trait autonomy (Spearman’s *r*(705) = −0.04, *p* = 0.23). The same holds for treatment-group-specific correlations of the bidirectional differences with autonomy (see Appendix A).

More importantly, however, absolute differences between pre- and post-intervention ratings across all seven items did correlate significantly negatively with trait autonomy (Spearman’s *r*(705) = −0.18, *p* < 0.01). This means that the more the rating shifted from pre- to post-intervention (regardless of the direction of shift), the lower the trait autonomy scores. Treatment-group-specific correlations are provided in Table 2. The negative correlation is most pronounced for the social worker with both the authoritarian message and the moralizing message.

The linear regression analysis tested these variations for statistical significance. Results showed that there was no overall effect of trait autonomy in predicting absolute pre-post differences (*b* = −0.04, 95% CI [−0.25, 0.16], *t* = −0.42, *p* = 0.68). However, sender was a significant predictor (*b* = 1.23, 95% CI [0.20, 2.27], *t* = 2.34, *p* = 0.02), as was the interaction of sender x autonomy (*b* = −0.33, 95% CI [−0.61, −0.04], *t* = −2.37, *p* = 0.03). Table 2 reveals the source of this interaction effect. The pre-post rating shifts were antagonized by autonomy more strongly in the social worker treatment group than in the state secretary treatment group. The interaction of sender x message (control) × autonomy was marginally significant (*b* = 0.38, 95% CI [−0.04, 0.79], *t* = 1.792, *p* = 0.07), suggesting that the group of participants receiving the authoritarian message from the high-authority figure showing a correlation of *r* = 0.08 (see Table 2) deviated slightly from the other treatment groups showing negative correlations between −0.12 and −0.30. Message framing or any of the other interactions showed no significant predictions (see Appendix A for the full regression table). Together, the predictors explain a small, but significant, portion of variance (*R^2^* = 0.041, 95% CI [0.01, 0.06], *F*(1, 695) = 5.37, *p* < 0.002).

### 3.3. Analysis 3: Reanalyses Controlling for Ceiling Effects

All items with pre-intervention ratings of 5 were eliminated from these analyses, excluding a total of 2768, out of 4949 pre-intervention ratings (56%) (see Appendix A S4 for item specific sample size with correction of ceiling effect).

Reanalysis 1. Again, the main ANOVA found only a significant effect of the repeated measures factor. The average post-intervention rating across all seven items (*M* = 3.27, *SD* = 0.96, *n* = 656) was significantly higher than the pre-intervention rating across all seven items (*M* = 2.79, *SD* = 0.85, *n* = 656; *F*(1, 650) = 1048.60, *p* < 0.001, *η^2^* = 0.41), this time with a large effect size. Message, sender, and all of the interactions did not show any significant effects. Message, sender, and interactions were not significant.

Reanalysis 2. Trait autonomy scores did not correlate significantly with bidirectional pre-post differences across all seven items (Spearman’s *r*(652) = 0.01, *p* = 0.71) but did correlate marginally significantly with the absolute pre-post differences across all seven items (Spearman’s *r*(652) = −0.07, *p* = 0.06). Treatment-specific correlations were not significant (Appendix A). The linear regression showed no significant interactions in this reanalysis (see Appendix A for the full regression table).

## 4. Discussion

Situated in the early months of the COVID-19 pandemic in Germany, we assessed common approval of health and safety regulations ordered by the government. We experimentally varied the framing and the sender of a fictitious social media post on Twitter promoting the regulations. We asked, firstly, whether an authoritarian sender and authoritarian framing would increase approval ratings (compared to moral and neutral control variants, respectively) and, secondly, whether this relationship would interact with trait autonomy of the recipients. In the spirit of open science, all our analyses were conducted as preregistered, and additional analyses are presented as exploratory analyses and Appendix A.

Across all treatment groups (i.e., all experimental manipulations of message framing and sender) and averaged across all seven items, we found that the Twitter messages significantly increased endorsement of the rules. However, despite being significant due to the large sample size, the effect was very small on average, explaining only 0.2% of the variance. This was caused in part because many of the ratings actually decreased from before to after the intervention, to our surprise; we thought at first that this may have been due to reactance effects in response to some items, especially those mentioning the “home”, namely items 1, 3, and 4 (see Appendix A for detailed graphs per item).

Another reason for the small size of the increase from before to after the intervention was the obvious presence of ceiling effects. This was not entirely unexpected (see preregistration, Analysis 3) as the same had been observed in prior studies investigating moral message framing on behavior during the COVID-19 pandemic where intervention effects are too small to pass the conventional levels of statistical significance [25]. During the early pandemic, when this survey was conducted, people were highly concerned and therefore willing to invest quite a lot into preventing the spread of the disease, as our findings showed. In many cases, their investments qualify as prosocial acts, maximizing joint welfare in the terminology of the social value orientation (SVO) framework [26], because they serve to protect the welfare of all, including oneself and others. Some measures, however, like wearing a mask, are more of an altruistic sacrifice whose purpose is merely to protect others [27,28]. We find it reassuring and praiseworthy that so many participants endorsed these regulation measures in a situation that was new to everyone, while protection measures severely restrained private rights and personal autonomy to a high degree. Participants of our study even reported more concerns for others than they did for themselves. Further promotion of this attitude via social media messaging was simply not needed for most measures (except for the new advice of mask-wearing).

In a statistical sense, the high level of endorsement was a problem because ceiling effects dampen the upward effects of experimental manipulations on prosocial/moral choice [29] and elsewhere. In fact, we did not observe any significant effects of our experimental manipulations in the analyses that did not consider individual differences in autonomy. To account for the problem, we reanalyzed the data in two ways. First, we eliminated all items with the maximum rating of 5 in the pre-intervention measurement and ran all analyses again. Second, we looked into the one item that appeared to show no ceiling effect (Item 5). This item referred to “wearing a mask in public indoor spaces”, which at the time of the survey had been completely voluntary; the official policy was still that there is not enough evidence to prove that wearing a mask significantly reduces a healthy person’s risk of infection, and the World Health Organization (WHO) presumed that wearing a mask might even create a false sense of safety and therefore lead to neglecting other hygiene measures [28,30].

In both reanalyses, we found large pre-to-post increases. First, averaged across all groups and across all seven items in the subset of data in which the ceiling effect had been statistically minimized, the pre-post measurement effect explained 41% of the variance (formerly 0.2%). Second, in Item 5, the pre-post measurement effect was also highly significant and went into the expected direction in all groups. The same was true for all other single items with statistically controlled ceiling effects (see Appendix A Appendix A and Appendix A). In addition, no seemingly “reactant” behavioral pattern was observed anymore in any of the groups. 

However, despite this clearly positive impact of the intervention overall, our experimental manipulations failed to show any statistically significant effects. Therefore, we conclude that the effects of the message framing and of the authority status of the sender were not significant in our sample, independent of any potentially dampening ceiling effects. 

What we did find though were significant effects of self-reported trait autonomy, and interactions of trait autonomy with the experimental manipulations. Across all groups, autonomy scores correlated negatively with the absolute pre-to-post intervention differences, meaning that the higher participants’ autonomy, the less they shifted upwards or downwards in their decision-making between the two rating measures. In other words, individuals high in trait autonomy resisted changing their ratings after reading the message more than those low in autonomy. 

Conversely, individuals low in autonomy shifted in their ratings more than those high in autonomy, both in accordance with the message and in opposition to it. This pattern is consistent with the idea of reactive autonomy, which describes autonomy as nonconformist resistance against social influences [15,16,17,18]. However, the relationship was larger in response to the social worker’s message than in the case of the state secretary, as suggested by the significant negative interaction of trait autonomy and sender (social worker) in the regression analysis. Perhaps this condition felt lenient enough to let individuals low in autonomy allow themselves to vary their decision-making, while those high in autonomy tended to stick to their prior ratings. The authoritarian sender, by contrast, led to more uniform decision-making across all participants, regardless of trait autonomy. 

In other words, the social worker, compared to the state secretary, may have increased diversity in opinion shifting, especially in the “downward” direction because the correlations were weaker for the bidirectional pre-post difference than for the absolute pre-post difference. Our manipulation checks indicated that the social worker was seen as more moralistic and more trustworthy than the state secretary. Trust and source credibility have been found before to enhance the effects of health-promoting messages in the context of the COVID-19 pandemic [31,32,33]. In our case, however, the trustworthiness and morality of the sender did not increase overall endorsement to the items, but they did interact with autonomy by unleashing higher variation in pre-to-post rating shifts, perhaps due to the involvement of positive emotions and the reduction in fear [32,33,34].

While we have focused on status, authoritarianism, and autonomy in the present study, we note that other features of sender, message, and recipient are likely to interact in determining the effects of health communication. In particular, the recipients’ sociodemographics and personality traits, other than reactive autonomy, influence perception and receptivity. One of the most important variables may be age. Older adults might prefer information through newspapers and national evening television, whereas young people prefer information through social media [35]. Further, it is likely that more authoritarian message framings and sender status can be found in the former, whereas the social media typically address recipients in a more colloquial way. This is no unidirectional relationship because recipients choose their sources, and sources in turn shape the communication preferences (and communication skills) of recipients. Eventually, the match between recipient, sender, and message framing may be the most crucial factor. The present study has selected only a fraction of the variables that can be considered when communication efficiency is sought to be optimized by more tailored approaches. 

One major limitation of our study is the problem of generalizability. First, the study was conducted in only one of the so-called WEIRD countries, namely Germany. For autonomy in particular, the cultural dimension of individualism may play a formative role [36]. However, differences in vertical versus hierarchical orientation can also have a major influence on submission to authorities. At present, our findings are indifferent to such variation and need comparison with different societal and cultural contexts. Second, the study was conducted in the early days of the pandemic, and it was still an extreme situation for most people, which might have influenced the generally high approval rates for the regulations. We would presume that a higher degree of uncertainty makes inter-individual differences in autonomy even more influential, as in the case of Item 5. 

The major strength of this study might be the new approach to measuring reactive autonomy using an experimental measure in conjunction with a self-report measure. In past experimentally oriented studies, researchers often struggled to make autonomy, in the sense of resistance to external or internal influences, measurable [37]. The challenge goes back to the complexity of defining autonomy in a uniform way: the concept of reactive autonomy [14,15,16] and its relation to self-regulative, reflective components of autonomy [17,18]. Additionally, from a feminist perspective, autonomy can be complemented by communion [38], which could be especially insightful in prosocial contexts like COVID-19 social distancing measures. 

The present study contributes a new approach to evaluating autonomy by focusing on its merely reactive meaning as resistance to external influences (in this case, social media communication). Beyond that, developmental differences during the lifespan, the motivational background of autonomous decision-making, and differences between cultures or societies are to be illuminated in future research. Furthermore, internal influences such as motivation and emotions (e.g., guilt/shame) might play a role in health communication where protecting others from the disease and also fear of getting infected oneself might drive attitudes and behaviors. 

Methodologically, our results can help to improve future surveys on related issues. First, the intervention effects were small—a problem that has also appeared in prior moral messaging studies during the COVID-19 pandemic [25,27]. Using a visual analog scale (VAS) to measure behavior instead of a Likert scale will increase the resolution of the dependent measure and might help to prevent ceiling effects. In addition, transfer effects might be reduced, as it is easier to remember a number between 1 and 5 from pre- to post-intervention rating than a detailed position on a VAS.

Second, the informative results we obtained with Item 5 about wearing a mask suggest that future research should not only ask about measures that are prescribed by officials but should instead focus more on protective measures that people may still be unsure about. For low-autonomy individuals in particular, it might be difficult to form and express an opinion that challenges official directives that are already implemented. As in every area of decision-making research, uncertainty and ambiguity enhance the person-specific component of the decision-making process [39], and based on the present findings, we can add trust as an additional variable for social settings. Under prosocial premises, participants may be more willing and more able to develop and report large or small changes in their opinions. 

## 5. Conclusions

The endorsement of health and safety regulations to protect against COVID-19 is generally high. Supporting public health communication via social media appears to have the strongest effect when there is some uncertainty about the effectiveness of the regulated behavior. Autonomous individuals tend to show more consistent endorsement of the regulations, whereas those low in autonomy allow their ratings to vary more in response to social messaging, especially when the sender has a nonauthoritarian social status and is trusted more. Disputed regulation measures are most susceptible to messaging interventions and their interactions with individual differences in autonomy. Future studies can build on these results in designing custom-tailored health communication to maximize its efficiency.

## Figures and Tables

**Figure 1 ijerph-18-07740-f001:**
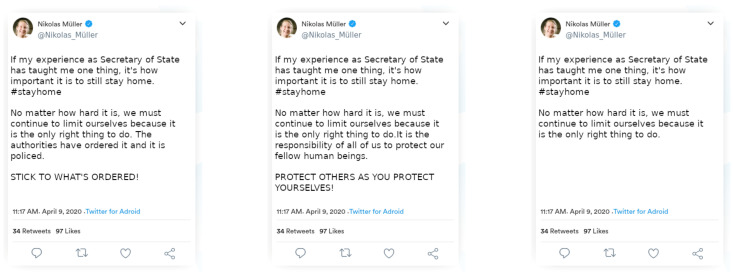
Twitter messages (translated to English) in three framings (from left to right: authoritarian message framing, moral message framing, and neutral message framing for control) sent by the state secretary (authoritarian sender); the social worker had the same photograph.

**Figure 2 ijerph-18-07740-f002:**
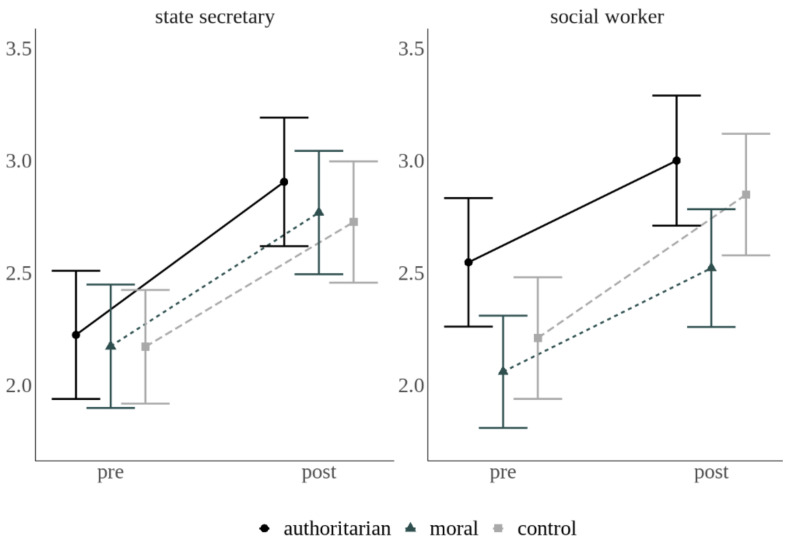
Mean ratings (95% CI) in response to Item 5, asking about wearing a mask in public indoor spaces, before (pre) and after (post) the message intervention.

**Table 1 ijerph-18-07740-t001:** Behavioral pre- and post-intervention measures as presented in the survey.

Item	
1	I reduce contact with other people outside the apartment to an absolute minimum.
2	I keep a minimum distance of 1.5 m to other people in public wherever possible.
3	I only spend time in public alone, with members of my household, or with one other person.
4	There are only very limited reasons for me to leave the house: emergency care, important purchases, doctor’s visit, necessary work, meetings, exams, sport, physical activity.
5	I wear a protective mask when I am in other indoor rooms.
6	For as long as schools and kindergartens are closed, I prevent my children from having any contact, or I would do this if I had children.
7	I abstain from personal contact with older relatives and persons at risk.

**Table 2 ijerph-18-07740-t002:** Spearman correlations between trait autonomy and absolute pre-post difference across all seven items for the 3 × 2 treatment groups.

	High Authority: State Secretary	Low Authority: Social Worker
authoritarian	*r* = 0.08 (116), *p* = 0.41	*r* = −0.25 (117), *p* = 0.01
moral	*r* = −0.15 (121), *p* = 0.10	*r* = −0.30 (117), *p* < 0.01
control	*r* = −0.15 (117), *p* = 0.11	*r* = −0.12(119), *p* = 0.19

Note. *p*-values are Holm adjusted for multiple tests.

## Data Availability

The data presented in this study are openly available in the Open Science Framework at https://doi.org/10.17605/OSF.IO/25GSN (accessed on 19 July 2021).

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
