# Peer review of "Effects of Message Framing, Sender Authority, and Recipients’ Self-Reported Trait Autonomy on Endorsement of Health and Safety Measures during the Early COVID-19 Pandemic"

_ijerph, 2021, doi:10.3390/ijerph18157740_

Round 1

Reviewer 1 Report

Dear authors

Overall, I liked this paper a lot. It is a well-written and well-structured paper which elaborates on social marketing and health related communications in response to the Covid pandemic. I provide below some minor revisions comments for specific sections which I hope will help you to elaborate further on specific arguments and improve the paper:

  • I notice that the introduction also acts as a literature review. I will strongly recommend to create a shorter version of the Introduction (with existing text) and create a small sub-section titled literature review. Accordingly, in the literature review you need to elaborate further on some of the key terms that you have been exploring. Firstly please elaborate on the study and findings of Tversky and Kahnemman and avoid the assumption that all readers will be familiar with the study and the findings. What did they study (sample) and how can their key findings be summarized?
  • Secondly, do you identify any limitations in these findings? In different cultures, societies and civilizations the sender and recipients of messages can come from diverse socio-cultural backgrounds and use different means of communications. Can we assume that a generalized approach can be applicable in different parts of the world? You mention that not all people are the same but you have to elaborate on this.
  • Finally, elaborate further on the Covid pandemic and avoid generalizations like ‘most countries around the world’. You can refer to continents or groups of countries if you wish but the response to Covid and marketing communications have been very diverse across the globe.
  • Equally, you can create a small-section titled (for example) ‘The case in Germany’ and highlight further the specific aspects of your context.

In general more work is required in the lit review to clarify your main hypothesis and research objective.

  • The methodology section is well-structured and the presentations of the findings are at high level. I will invite you to include specific citations instead of hyperlinks to studies. The fictitious post is imaginative.
  • I am afraid the problem of lacking an in-depth analysis of the literature review also moves to the discussion. I can see some of the practical applications and viability of your study, however in which academic area do these findings seek to make a contribution? Which future studies (and in which discipline) can benefit from this paper?
  • How can we generalize these findings in other countries and if so where and why?

I do believe that more analysis is required in the Discussion section so as to show the contribution. Overall, this is a strong paper in terms of sample and methodology but both the lit review and Discussion should be further developed to highlight your contribution.

Author Response

Dear Editor,

Dear Reviewers,

please see PDF attached.

Reviewer 2 Report

This is important research in particular for designing successful interventions in the COVID-19 pandemic. As recognized by the authors human solidarity and prosocial behavior play a crucial role to meet this big societal challenge.

The introduction is well written, the design is well balanced.

I have concerns with the method and result sections concerning the questionnaire measuring trait autonomy. Empirically, autonomy orientation is associated positively with self-evaluation, self-awareness, self-actualization and ego development but also with higher moral reasoning. The autonomy questionnaire contains items of different other questionnaires and only one item with thoughts about not only the respective individual but also others: “Most of the time I can tell how my actions are going to affect others”. The short questionnaire might contain different dimensions, it is not proved and no reliability measure is provided.

The sample is not well described, but called representative – how was it ensured. What were the two attention check questions?

The sample sociographic variables are not included in the analyses. It might make a big difference if older people who were identified as vulnerable group – also officially – or younger ones who felt not at risk rated their prevention behavior during the pandemic. Additional to age, gender and education might make a difference.

The analyses and results are not very well presented and should be checked in details and some data should be reanalyzed.

Page 6 210-217 – it is not clear which results are related to which items? Please provide the numbers of items (measurement).

E.g considering item 5 is indicated as the only item without ceiling effect in the paper, in the supplements the ANOVAS with or without ceiling effect contain either 701 individuals or without ceiling effect less (about 147 to 343 individuals). Even item 5 “wearing masks” shows a ceiling effect.

The description of the tables and figures are not clear. The regression analyses for example in the supplement indicate b represents unstandarized regression weights, but only ß are represented.

To include the sociodemographic variables hierarchical regression analyses could be calculated. As a first step the sociodemographic variables could be used as predictors, and as a second step the significant sociodemographic variables and the experimental variables could be used as predictors for the criteria measurements (the 7 items or the mean of the seven items).

It is well known that tailored interventions and communication in health issues like COVID-19 pandemic including the characteristics of the recipients might be more effective than only address the kind of senders and the kind of messages.

Author Response

(The authors gave the same response as above.)

Round 2

Reviewer 2 Report

The authors considered all aspects of the fisrt review